# Nutritional and Biochemical Outcomes After Total Versus Subtotal Gastrectomy: Insights into Early Postoperative Prognosis

**DOI:** 10.3390/nu17132146

**Published:** 2025-06-27

**Authors:** Fawzy Akad, Cristinel Ionel Stan, Florin Zugun-Eloae, Sorin Nicolae Peiu, Nada Akad, Dragos-Valentin Crauciuc, Marius Constantin Moraru, Cosmin Gabriel Popa, Liviu-Ciprian Gavril, Roxana-Florentina Sufaru, Cristina Preda, Veronica Mocanu

**Affiliations:** 1Department of Morpho-Functional Sciences I, “Grigore T. Popa” University of Medicine and Pharmacy, 700115 Iasi, Romania; fawzy_akad@umfiasi.ro (F.A.); eloae.zugun@umfiasi.ro (F.Z.-E.); dragos.crauciuc@umfiasi.ro (D.-V.C.); marius.moraru@umfiasi.ro (M.C.M.); cosmin-gabriel.popa@umfiasi.ro (C.G.P.); liviu.gavril@umfiasi.ro (L.-C.G.); roxana-florentina.sufaru@umfiasi.ro (R.-F.S.); 2Center for Obesity BioBehavioral Experimental Research, “Grigore T. Popa” University of Medicine and Pharmacy, 700115 Iasi, Romania; nada_akad@umfiasi.ro; 3Transcend Research Center, Regional Oncology Institute, 700483 Iasi, Romania; 4Department of Vascular Surgery, “Grigore T. Popa” University of Medicine and Pharmacy, 700115 Iasi, Romania; sorin-nicolae.peiu@umfiasi.ro; 5Department of Medical Sciences II, “Grigore T. Popa” University of Medicine and Pharmacy, 700115 Iasi, Romania; cristina.preda@umfiasi.ro; 6Department of Morpho-Functional Sciences II, “Grigore T. Popa” University of Medicine and Pharmacy, 700115 Iasi, Romania

**Keywords:** gastric cancer (GC), gastrectomy, carcinoembryonic antigen (CEA), carbohydrate antigen 19-9 (CA19-9), prognostic factors

## Abstract

Gastric cancer remains a significant global health burden, with curative treatment relying on surgical resection, typically total or subtotal gastrectomy. However, the procedure frequently triggers acute metabolic and nutritional disturbances that may impact recovery. **Objective**: This prospective study aimed to investigate whether the type of gastrectomy (total vs. subtotal) influences early postoperative biochemical and hematological alterations, with particular attention to nutritional impact. **Methods**: A cohort of 295 patients (123 female, 172 male) who underwent gastrectomy for gastric cancer at the Institute of Oncology Iași (2023–2024) was evaluated. Laboratory parameters, including hemoglobin, hematocrit, lymphocyte and platelet counts, serum albumin, total protein, sodium, potassium, creatinine, and urea, were analyzed preoperatively and on postoperative day 14 using standard clinical methods. **Results**: Anemia was observed in over 90% of patients, irrespective of sex or procedure type. Electrolyte imbalances (notably hyponatremia and hypokalemia) and indicators of nutritional deficit (hypoalbuminemia, low creatinine) were highly prevalent, with a greater frequency among female patients. Total gastrectomy was associated with more severe biochemical and nutritional alterations compared to subtotal procedures. **Conclusions**: Total gastrectomy significantly exacerbates early postoperative metabolic and nutritional derangements. These findings reinforce the need for proactive, personalized postoperative nutritional and electrolyte management strategies to support recovery and reduce complication risks.

## 1. Introduction

Gastric cancer (GC) is a biologically diverse and highly aggressive malignancy, currently ranking as the fourth most common cause of cancer-related mortality worldwide. GC represents a major global health burden, with over 1 million new cases and approximately 770,000 deaths reported annually, according to GLOBOCAN 2020 data [1]. It is the fifth most commonly diagnosed cancer and the fourth leading cause of cancer-related mortality worldwide. The incidence varies by region, with the highest rates observed in East Asia, Eastern Europe, and parts of South America. Despite advancements in surgical techniques, chemotherapy, and targeted therapies, the overall prognosis for GC remains poor, particularly in patients diagnosed at advanced stages. Risk factors include Helicobacter pylori infection, dietary habits, smoking, and genetic predisposition. Multimodal approaches are now standard in locally advanced disease, but outcomes are still largely dependent on the timing of diagnosis and the physiological status of the patient [1].

In this study, we aimed to compare early postoperative biochemical and nutritional changes between total and subtotal gastrectomy in patients with gastric cancer, focusing on the acute postoperative phase (up to 14 days). The goal was to determine whether the extent of gastric resection leads to distinct metabolic and hematological alterations relevant to nutritional status, thereby supporting the need for tailored postoperative monitoring and intervention strategies.

Due to its subtle clinical presentation and frequently asymptomatic early stages, GC is often diagnosed at an advanced stage, complicating curative interventions and contributing to elevated recurrence rates and unfavorable long-term prognosis [2].

Serum tumor markers, either produced directly by tumor cells or induced in response to the presence of neoplastic tissue, serve as valuable indicators of tumor burden and biological behavior [3]. These biomarkers have been widely employed in GC for early detection, monitoring for recurrence after curative resection, and assessing prognosis [4].

Among the most studied biomarkers, carcinoembryonic antigen (CEA) and carbohydrate antigen 19-9 (CA19-9) have been proposed as potential prognostic indicators in GC [5,6,7].

Tumor markers reflect underlying genetic, molecular, and cellular changes driven by malignancy, and they are widely used not only for diagnostic purposes but also for disease surveillance and evaluation of treatment efficacy [4,8,9].

Malnutrition is a common and clinically relevant concern among patients with gastric cancer, affecting up to 80% of cases at various disease stages. This condition arises from a combination of reduced oral intake due to tumor location and dysphagia, increased catabolism driven by systemic inflammation, and side effects of treatments such as chemotherapy or surgery. Malnourished patients are at greater risk for postoperative complications, including impaired wound healing, infectious morbidity, and prolonged hospitalization. Several studies have also demonstrated that nutritional deficits independently predict poor long-term oncologic outcomes and decreased survival. Therefore, early recognition and correction of metabolic and nutritional imbalances are essential components of perioperative care in gastric cancer management [10,11].

The type and extent of surgical resection play a central role in determining the degree of postoperative metabolic stress and nutritional compromise. Total gastrectomy, which entails the complete removal of the stomach, results in the loss of gastric reservoir function, intrinsic factor secretion, and critical hormones involved in appetite and digestion. These physiological losses contribute to significant postoperative challenges, including malabsorption, rapid weight loss, anemia, and hypoalbuminemia. In contrast, subtotal gastrectomy typically preserves more digestive capacity, leading to relatively less severe disturbances. The degree of postoperative nutritional impairment is thus closely linked to surgical extent, making it essential to evaluate these changes in relation to the surgical approach [12,13].

Although the long-term nutritional and metabolic consequences of gastrectomy have been widely studied, limited data exist regarding the acute-phase biochemical alterations occurring in the first two postoperative weeks. Most existing studies focus on survival or complication rates but do not systematically analyze how early postoperative laboratory profiles evolve depending on the type of surgery. This gap in knowledge limits our ability to implement early, targeted nutritional interventions. By comparing total and subtotal gastrectomy outcomes in a prospective design, the current study aims to identify short-term hematological and biochemical patterns with potential prognostic significance and to support the development of more personalized postoperative care strategies [14].

## 2. Materials and Methods

In this study, we conducted a comprehensive analysis of the longitudinal evolution of key biochemical and hematological parameters in a cohort of patients who underwent total gastrectomy (TG) and subtotal gastrectomy (STG) (*n* = 295).

The patients were selected over the last two years (2023–2024) and were monitored at the Institute of Oncology Iași, where the surgical intervention was also performed. The diagnosis of gastric cancer was histologically confirmed by endoscopic biopsy. Tumor staging was performed using contrast-enhanced thoraco-abdominal computed tomography (CT), which enabled precise assessment of tumor location, depth of invasion, and regional lymph node involvement. All cases were reviewed in a multidisciplinary tumor board composed of surgeons, oncologists, radiologists, and nutritionists to determine the optimal therapeutic strategy. Only patients who were eligible for curative resection and did not receive neoadjuvant chemotherapy were included in order to avoid confounding the postoperative biochemical and nutritional data under analysis. Candidates for upfront surgery were selected based on clinical stage T1–T3N0–N1 without evidence of distant metastasis, good performance status (ECOG 0–1), and absence of major comorbidities that would contraindicate immediate surgical intervention. The type of gastrectomy was selected in accordance with established surgical guidelines, based primarily on preoperative tumor location, extent of local invasion, and oncologic safety margins. Tumor localization was determined preoperatively through upper gastrointestinal endoscopy and CT imaging. Accordingly, total gastrectomy (TG) was indicated for proximally located or diffusely infiltrative tumors, while subtotal gastrectomy (STG) was performed for tumors confined to the distal part of the stomach (Table 1) [15,16].

Postoperative follow-up included histopathological evaluation of the resected specimens, with full pTNM staging, assessment of tumor differentiation, and identification of lymphovascular or perineural invasion. Based on these findings and in accordance with international guidelines (ESMO, NCCN), patients presenting with stage II or III disease, nodal involvement, or other high-risk pathological features were recommended for adjuvant chemotherapy [15,17]. These decisions were made by a multidisciplinary tumor board, and eligible patients were referred to oncology for individualized treatment planning. Patients undergoing adjuvant therapy were further monitored by oncologists through a structured follow-up protocol that included periodic clinical evaluations, laboratory tests, nutritional status monitoring, imaging (such as contrast-enhanced CT every 6 months), and tumor marker assessments (CEA and CA 19-9) to ensure oncologic control.

The primary focus of the study was to assess the impact of the surgical technique, TG versus STG, on postoperative biochemical and hematological parameters. Lymph node status and local tumor invasion were taken into account when determining the type of surgical intervention, in accordance with oncological practice. These factors were not directly analyzed in the scope of this study. This analysis aims to contribute to a better understanding of the physiological response following different extents of gastric resection, supporting more informed perioperative management.

Histopathological analysis was based on the World Health Organization (WHO) classification of gastric adenocarcinoma, which defines the following subtypes, each influencing surgical approach and prognosis:

Tubular Adenocarcinoma: Well-organized glandular structures, varying differentiation (G1–G3), influencing the extent of lymph node dissection and resection margins.

Papillary Adenocarcinoma: Papillary projections into the lumen are often associated with higher lymphatic invasion, necessitating more extensive lymphadenectomy.

Mucinous Adenocarcinoma: ≥50% extracellular mucus, correlated with poor prognosis and peritoneal dissemination, frequently requiring TG.

Signet-Ring Cell Adenocarcinoma: Characterized by poorly cohesive tumor cells, early peritoneal spread, and diffuse invasion, often necessitating TG with D2 lymphadenectomy.

Mixed-Type Adenocarcinoma: Combination of intestinal and diffuse components, influencing the choice between subtotal or TG based on the dominant histological pattern. Globally, tubular adenocarcinoma accounts for approximately 40–50% of gastric cancer cases, papillary adenocarcinoma for 5–10%, mucinous adenocarcinoma for 3–7%, signet-ring cell carcinoma for 15–25%, and mixed-type adenocarcinoma for 10–15%, according to WHO classifications and large-scale histopathological studies [18,19,20,21].

By establishing these associations, this section aims to clarify how tumor burden, lymph node metastasis, and histopathological characteristics guide the selection of the most indicated surgical approach, ultimately influencing patient outcomes and recurrence risks.

The analysis of surgical type according to gender showed that among female patients (*n* = 122), 65 underwent STG (53.3%) and 57 TG (46.7%), whereas among male patients (*n* = 172), 80 underwent STG (46.5%) and 92 TG (53.5%). This indicates a slightly higher tendency towards TG among male patients compared to female patients.

### 2.1. Baseline and Postoperative Dynamics of Biochemical and Nutritional Markers in GC Surgery

Peripheral blood samples were taken before surgical intervention and two weeks postoperatively under rigorous and standardized conditions to ensure the uniformity and accuracy of the results. Sampling was performed preoperatively after a minimum fasting period of 8 h to minimize external influences on metabolic and nutritional parameters. The collection took place 24 h before the surgical intervention, a time point deemed optimal for capturing the patients’ baseline biochemical and hematological status. All procedures adhered to good laboratory practice guidelines, utilizing sterile materials and standardized techniques to prevent contamination or sample degradation.

### 2.2. Evaluation of Biochemical Profiles

The analysis encompassed a comprehensive panel of biochemical and hematological parameters, selected to provide an in-depth evaluation of the patient’s metabolic and nutritional status. Among the assessed markers were serum albumin, prealbumin, sodium, potassium, creatinine, urea, hemoglobin, hematocrit, lymphocytes, and platelets. These parameters were chosen for their relevance in assessing nutritional status, systemic inflammatory response, renal function, hematological balance, and tumor burden (Table 2).

Data distribution was assessed using the Kolmogorov–Smirnov and Shapiro–Wilk tests. None of the biochemical parameters followed a normal distribution. To compare the two surgical groups (TG vs. STG), the Mann–Whitney U test was used for nonparametric variables, including biochemical markers, tumor markers, and nutritional parameters, both preoperatively and at 14-day follow-up. Statistical analyses were performed using SPSS version 26.0, with a significance level set at *p* < 0.05. The results highlight the relevance of integrating oncological and biochemical evaluations in selecting the optimal surgical approach and improving perioperative patient care.

## 3. Results

### 3.1. Patient Characteristics

In this study, we analyzed a cohort of *n* = 295 patients who underwent TG or STG at the Institute of Oncology Iași during 2023–2024. The demographic characteristics of the study population provide valuable insights into the patient profile and potential influencing factors. The mean age of the patients was 64.96 years (±10.45 SD), with a median age of 66 years and an age range between 31 and 85 years, suggesting that GC predominantly affects elderly individuals, consistent with existing epidemiological data (Figure 1). Regarding sex distribution, 59% of the patients were male (*n* = 174) and 41% were female (*n* = 121), reaffirming the higher prevalence of GC in men. Concerning the living environment, 56% of the patients (*n* = 164) resided in urban areas, while 44% (*n* = 131) were from rural regions, indicating a relatively balanced distribution between urban and rural settings.

In this study, we aimed to evaluate the clinicopathological and biochemical determinants influencing the extent of gastrectomy, comparing TG, *n* = 150 and STG, *n* = 145 in a cohort of *n* = 295 patients.

The primary objective of this study was to analyze the biochemical and hematological changes associated with two types of surgical intervention for GC, in accordance with current clinical guidelines. Surgical approach was selected based on established oncological criteria, including lymph node involvement, histopathological tumor grading according to WHO classification, and the presence of loco-regional invasion. Understanding how surgical extent influences physiological responses is essential for optimizing perioperative management and improving patient outcomes.

Regarding local invasion, the cohort was composed of two distinct subgroups: 133 patients (45.1%) showed no evidence of loco-regional tumor infiltration, while 162 patients (54.9%) presented with confirmed local invasion at diagnosis. This distribution suggests a slight predominance of invasive cases, supporting the advanced presentation commonly observed in GC.

For the number of positive lymph nodes, values ranged broadly from 0 to 67, indicating considerable variability in nodal involvement. A substantial portion of patients (25.8%) had no lymph node metastases, while the remaining exhibited varying degrees of lymphatic dissemination. Although most patients clustered in the lower range (1–10 positive nodes), a smaller subgroup displayed extensive nodal spread, with values exceeding 30 in a few cases, highlighting the biological heterogeneity of GC and its diverse metastatic potential.

To assess differences between the two groups, a Mann-Whitney U test was performed for each variable. The analysis revealed that the number of positive lymph nodes (*p* = 0.01) and loco-regional invasion (*p* = 0.001) demonstrated statistically significant differences between patients undergoing TG versus STG. Patients with a higher burden of lymphatic metastases and extensive local invasion were significantly more likely to undergo TG, reinforcing the idea that tumor dissemination plays a pivotal role in the extent of surgical resection. The mean rank for positive lymph nodes was (M = 135.3) in the STG group and (M = 160.2) in the TG group, indicating a higher metastatic burden in patients requiring more extensive resection. Similarly, patients presenting with local tumor invasion were more frequently associated with TG. The mean rank in the TG group was 161.40, compared to 134.14 in the subtotal group, indicating a greater burden of loco-regional disease in those requiring a more extensive surgical approach. The statistical analysis yielded a Mann-Whitney U value of 8.86 and a *p*-value of 0.001, confirming a highly significant difference between the two groups. This reinforces the clinical observation that local invasion serves as a critical criterion in determining the extent of gastric resection, supporting TG as the preferred intervention.

Regarding histopathological tumor grading, based on the WHO classification, it revealed a varied distribution of adenocarcinoma subtypes. Tubular adenocarcinoma represented the largest group (*n* = 123, 41.7%), followed by papillary adenocarcinoma (*n* = 34, 11.5%), mucinous adenocarcinoma (*n* = 28, 9.5%), signet-ring cell adenocarcinoma (*n* = 87, 29.5%), and mixed-type adenocarcinoma (*n* = 23, 7.8%). The variability in these histopathological subtypes highlights the heterogeneity of GC within the studied cohort.

The results (*p* = 0.09) indicated a trend toward a greater prevalence of poorly differentiated tumors in the TG group, although the difference did not reach statistical significance. The grading system included subtypes such as tubular, papillary, mucinous, signet-ring cell adenocarcinomas, and mixed, each with varying degrees of differentiation and biological aggressiveness. While poorly differentiated tumors were more frequently observed in patients undergoing TG, the lack of statistical significance suggests that grading alone is not the primary determinant in surgical decision-making, but rather, it acts as a secondary factor in conjunction with lymph node metastasis and locoregional invasion.

Although both CEA and CA19-9 levels remained above the reference range, the continued decline observed indicates the absence of early recurrence and potential prognostic value in postoperative monitoring. These results reinforce the importance of serial measurements over time, rather than relying on single preoperative values, and support their role in postoperative surveillance as recommended by current oncologic guidelines.

### 3.2. Comparative Analysis of Biochemical Parameteres at Baseline and Follow-Up

Prior to surgical intervention, a baseline panel of biochemical, hematological, and nutritional parameters was obtained for all patients. These preoperative measurements revealed values largely within normal reference ranges across both STG and TG groups. No statistically or clinically significant differences were observed between the two groups at this stage, indicating a comparable physiological and nutritional status prior to surgery. This uniformity supports the premise that any subsequent metabolic or hematological alterations observed postoperatively can be attributed primarily to the type and extent of surgical intervention rather than pre-existing disparities. A summary of the preoperative values is provided in Table 3 and Table 4.

The postoperative dataset used in this study was collected on postoperative day 14, representing the most comprehensive and consistent time point for which laboratory data were available across the entire patient cohort. This early postoperative window was selected to capture acute physiological responses to STG and TG, including changes in hematologic profiles, electrolyte balance, protein metabolism, and nutritional indicators. Although longer-term follow-up measurements would offer greater insight into recovery patterns and sustained outcomes, such data were not uniformly available. Therefore, the analysis focused on this standardized early time point to allow for consistent intergroup comparison and to identify early postoperative trends associated with each surgical technique.

The presence of such hematological impairments highlights the increased risk of anemia following TG, likely due to more extensive blood loss, reduced iron absorption, and diminished intrinsic factor production affecting vitamin B12 metabolism. Significant differences were identified in electrolyte levels, with sodium (*p* = 0.005) and potassium (*p* = 0.02) being more affected in TG patients. These alterations may reflect postoperative fluid imbalances, altered renal excretion, and the physiological adaptations to total loss of gastric function, increasing the likelihood of electrolyte disturbances requiring close postoperative monitoring (Table 5 and Table 6) (Figure 2 and Figure 3).

Beyond these statistically significant findings, trends were also noted in platelet count and protein metabolism. The mean platelet count was lower in TG patients (*p* = 0.05), suggesting a possible association with thrombocytopenia, though not reaching statistical significance. Additionally, albumin (*p* = 0.1) and total protein (*p* = 0.2) levels tended to show more depletion in TG patients, reinforcing the known risk of protein-energy malnutrition following total gastric resection.

The observed decrease in albumin may be attributed to impaired protein digestion and absorption, which are exacerbated by the complete removal of the stomach and subsequent malabsorption of essential nutrients. A similar trend was observed in the prognostic nutritional index (PNI), which showed lower values in TG (*p* = 0.1), suggesting a greater nutritional burden postoperatively. While these findings were not statistically significant, they highlight the clinical relevance of nutritional interventions in TG patients.

Conversely, renal function markers such as creatinine (*p* = 0.1) and urea (*p* = 0.3) showed no significant differences between groups, indicating that, despite altered protein metabolism, renal impairment was not a distinguishing factor between TG and STG.

The comparative analysis between total and STG groups revealed significant differences in key hematological and biochemical parameters, reflecting the physiological burden associated with more extensive surgical resection. Patients undergoing TG demonstrated significantly lower hemoglobin and hematocrit levels (*p* = 0.01), as well as reduced platelet counts (*p* = 0.01), indicating a more pronounced hematologic impact. Additionally, electrolyte imbalances were more evident in the TG group, with sodium (*p* = 0.008) and potassium (*p* = 0.01) levels significantly lower compared to those who underwent subtotal resection, suggesting a heightened risk of fluid and electrolyte disturbances in this study. Although renal function markers such as creatinine and urea showed no statistically significant differences (*p* > 0.05), their variability highlights the potential influence of individual metabolic responses. Nutritional indicators, including serum albumin and the PNI, were also lower in the TG group, though without statistical significance (*p* > 0.05). These findings collectively underscore the greater metabolic and hematologic vulnerability of patients undergoing TG and support the implementation of tailored postoperative monitoring and early nutritional intervention in this group (Table 7) (Figure 4).

## 4. Discussion

Gastrectomy, whether total or subtotal, is a fundamental surgical intervention for the management of GC, particularly in cases of locally advanced disease. This procedure induces significant physiological alterations in the digestive system, often resulting in severe malnutrition due to anatomical and functional changes that adversely affect dietary intake, digestion, and nutrient absorption.

GC remains a highly aggressive and biologically diverse malignancy, with considerable variability in prognosis even within the same clinical stage. This heterogeneity underscores the need for personalized surgical strategies and accurate risk stratification [22].

The ongoing debate between subtotal and TG, particularly in distal GC, highlights the importance of balancing oncological radicality with surgical safety. Although TG is often perceived as more extensive, it may be associated with higher perioperative risk without necessarily offering superior long-term survival. In cases where oncologic outcomes are comparable, the surgical approach associated with lower morbidity may be preferable [23].

Given the high prevalence and aggressive nature of GC, current research highlights the pivotal role of surgery, especially gastrectomy, as the primary curative strategy for patients with advanced localized disease. However, the profound metabolic and nutritional implications of this intervention necessitate comprehensive perioperative management. The present study evaluated the clinical and biochemical factors associated with the extent of surgical resection in patients diagnosed with GC. Our findings underscore the critical role of nodal metastasis and locoregional tumor invasion in surgical decision-making. Specifically, the number of positive lymph nodes and the presence of local invasion emerged as the most robust predictive indicators for selecting TG, aligning with the findings of previous studies that emphasized the oncological burden as a central determinant of surgical extent [24,25]. Tumor markers such as CEA and CA 19-9 were measured postoperatively; however, their inclusion in the primary analysis was avoided, as the timing of assessment, only 14 days after surgery, was considered too early to yield clinically meaningful insights. While a decrease in marker levels was noted in some patients, the variation lacked statistical significance and was not sufficient to support robust conclusions regarding short-term treatment response or recurrence risk. Similar variability in the prognostic utility of these markers across different patient subgroups and tumor stages was also reported by Zhou et al. (2015) [26]. Although serum levels of CA19-9 and CEA are frequently elevated in gastrointestinal malignancies and have been correlated with tumor burden, their role as standalone determinants in preoperative decision-making remains limited [27,28]. Therefore, while included in our analysis for exploratory purposes, these markers should be interpreted with caution and integrated within a broader oncologic and clinical assessment.

The evaluation of biochemical and hematological parameters in patients undergoing GC surgery provided a valuable overview of their preoperative physiological status. This allowed for the early detection of nutritional deficiencies and inflammatory risk factors, which can be proactively addressed through tailored perioperative interventions. The preoperative data also served as a reliable baseline for assessing the physiological impact of surgical resection, supporting a more informed and individualized patient management strategy. Postoperative analysis revealed significant alterations in key metabolic indicators, particularly in patients undergoing TG. The loss of gastric secretory function was associated with notable disruptions in fluid and electrolyte homeostasis, including hyponatremia and hypokalemia, which may predispose patients to renal dysfunction and delayed recovery. These imbalances, along with hypoalbuminemia and hematological shifts, were found to adversely affect clinical trajectory. Understanding and addressing these metabolic consequences is essential for optimizing perioperative care, reducing morbidity, and improving long-term outcomes in GC patients.

Nutritional support during the 14-day postoperative period was delivered on an individual basis rather than through a formal protocol, but under the supervision of a clinical nutritionist. Electrolyte disturbances, most frequently hypokalemia, were corrected promptly with intravenous potassium replacement; mild to moderate hyponatremia received intravenous isotonic saline (0.9% sodium chloride solution) as part of fluid and electrolyte management, based on individual clinical and laboratory findings; and patients who exhibited clinically relevant hypoalbuminemia received targeted protein supplementation with human serum albumin solutions. These interventions were guided by daily laboratory monitoring and physician judgement, ensuring that overt ionic or protein deficits were addressed before discharge.

The analysis demonstrated distinct hematological and biochemical alterations associated with the extent of gastric resection, suggesting a more significant physiological burden in patients undergoing TG. Lower postoperative levels of hemoglobin, hematocrit, and platelet count in this group may reflect a more profound impact on hematopoietic balance and perioperative hemodynamics. Moreover, the observed disturbances in serum sodium and potassium levels indicate a disruption in electrolyte homeostasis, likely secondary to the loss of gastric regulatory functions and changes in fluid distribution following complete gastric removal. Although renal function markers remained within comparable ranges, interindividual variability suggests differences in metabolic adaptation and renal reserve. From a nutritional standpoint, the downward trends in serum albumin and prognostic nutritional index among patients with TG highlight an increased vulnerability to protein-energy depletion, even in the absence of overt clinical signs. These findings emphasize the need for proactive metabolic surveillance and early nutritional support strategies, particularly in patients undergoing more extensive resections, to mitigate the risk of postoperative complications and support optimal recovery.

Surgical trauma, particularly in oncologic procedures, induces significant immunological changes that can influence both short- and long-term outcomes. It is well-established that white blood cell (WBC) subsets undergo dynamic changes in peripheral blood following major surgery due to factors such as neutrophil survival and lymphocyte apoptosis. These changes, which reflect the patient’s immunologic status, are modulated by surgical stress, psychological factors, and tumor burden and have been proposed as prognostic markers in cancer populations. Given that cancer patients may already present with an altered immune profile, the immunosuppressive effect of surgery may further increase the risk of tumor progression or metastasis if immune competence is not restored promptly. Thus, understanding perioperative immune dynamics is essential, especially in the context of patient-specific variables such as sex [29,30,31].

While our study did not reveal statistically significant differences in biochemical or hematological markers between male and female patients, emerging evidence suggests that gender-related immunological responses to surgical stress do exist and may be clinically relevant. A study by Mi Sook Gwak et al. (2007) [32] found that female patients exhibited a more immunocompromised profile of WBC subsets in peripheral blood in the early postoperative period after gastrectomy for gastric cancer. The authors concluded that these transient immune differences may influence long-term outcomes, including tumor recurrence, and emphasized the need for further investigation. These findings, while not replicated in our dataset, highlight the possibility of subtle, sex-specific immune perturbations that may not be apparent in standard laboratory parameters but could manifest in functional immune shifts or clinical outcomes over time [32].

Sex-related differences in immune response may, in part, be mediated by hormonal influences. Testosterone and progesterone are generally associated with anti-inflammatory responses, whereas estrogen has a more complex role, exhibiting either pro- or anti-inflammatory effects depending on its concentration and the tissue microenvironment. Although these mechanisms were not directly assessed in our study, they offer a plausible biological explanation for the sex-related immune variability reported in the literature [33].

Our findings further highlight the increased biological burden associated with TG. Comparative analysis of biochemical parameters between patients undergoing total versus STG revealed significant postoperative differences in hemoglobin, hematocrit, sodium, and potassium levels, indicating a greater risk of anemia and electrolyte imbalance in the TG group. This observation is clinically relevant, as it underscores the necessity for intensified monitoring and proactive supportive care in patients subjected to more extensive resections. While the biochemical and hematological data analyzed in this study were collected during the early postoperative period, they were used to assess the immediate physiological impact of subtotal versus TG. This timeframe was selected to capture acute metabolic and nutritional alterations attributable to surgical stress, extent of resection, and early postoperative adaptation.

The previous literature supports the need for a nuanced approach to surgical planning in GC. Palaj et al. (2021) [24] emphasized that while STG may be the preferred approach for middle and distal-third tumors due to similar survival rates and lower complication risks, TG continues to be widely practiced, particularly in high-volume Western centers [24,34].

In parallel, there is a growing recognition of the prognostic value of hematological parameters in GC. Recent studies, such as the one by Feng et al. (2018), have demonstrated that low lymphocyte counts and high monocyte counts are associated with poorer outcomes in GC, with lymphocyte levels emerging as independent prognostic markers [35]. These findings resonate with our results, particularly in relation to lymphocyte and platelet-to-lymphocyte ratio (PLR), further supporting the utility of these inexpensive and accessible parameters in clinical risk stratification.

This study was designed to explore acute postoperative changes in biochemical and hematological parameters, with follow-up limited to the first 14 days after surgery. The rationale behind this approach was to isolate and examine the physiological impact of the type of gastrectomy performed, independent of each patient’s preexisting metabolic or nutritional status. By comparing postoperative outcomes between STG and TG groups at a fixed time point, we aimed to highlight interventional differences in nutritional burden and systemic recovery. This strategy allows for a clearer evaluation of how surgical extent alone influences early postoperative nutritional status, hematologic parameters, and metabolic stability, without the confounding effect of variable baseline conditions. Emphasis was placed particularly on the PNI, albumin levels, and electrolyte disturbances, as these are clinically relevant indicators of postoperative nutritional risk and are known to correlate with recovery trajectories and complication rates. While individual trajectories may provide valuable insight in longitudinal studies, our primary aim was to delineate the early impact of surgical strategy on systemic physiology, thus justifying the choice of a cross-sectional comparative model.

Although patients are, in theory, enrolled in outpatient surveillance following discharge, structured long-term follow-up beyond the hospital stay (e.g., at one or three months) was not included in the present study design, as a proportion of patients underwent adjuvant therapy after surgery, which could significantly alter hematological and biochemical profiles. To avoid such confounding factors, we chose to focus exclusively on the acute postoperative period in order to assess the direct impact of the surgical intervention on metabolic and nutritional parameters. We acknowledge this as a limitation, as the long-term trajectory of these early biochemical alterations remains unknown. Future prospective studies should incorporate extended follow-up periods in order to assess whether such disturbances persist, resolve, or worsen over time, and to enable a more robust correlation between early laboratory findings and long-term clinical outcomes.

## 5. Conclusions

The findings of this study demonstrate that the extent of gastric resection has a measurable impact on early postoperative biochemical and hematological profiles, with TG patients exhibiting more significant disturbances in hemoglobin, hematocrit, white blood cells, platelets, sodium, and potassium levels. These differences reflect a higher metabolic burden and increased risk of early postoperative complications in this group. The observed trends in albumin and protein levels, though not always statistically significant, further emphasize the need for vigilant nutritional and biochemical surveillance. These results support the implementation of specific postoperative care strategies, prioritizing early nutritional intervention and targeted correction of electrolyte and hematological imbalances, particularly in patients undergoing TG.

## Figures and Tables

**Figure 1 nutrients-17-02146-f001:**
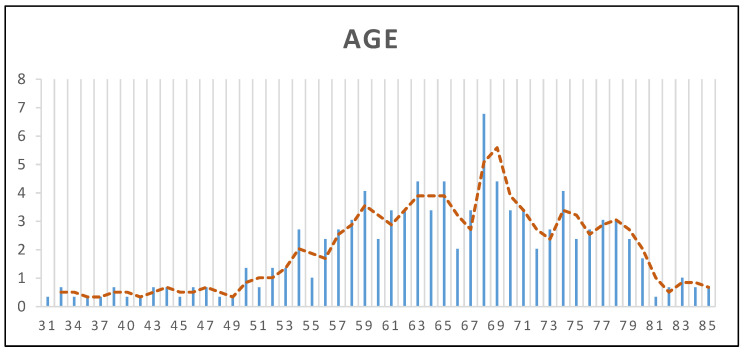
Age distribution of the study population.

**Figure 2 nutrients-17-02146-f002:**
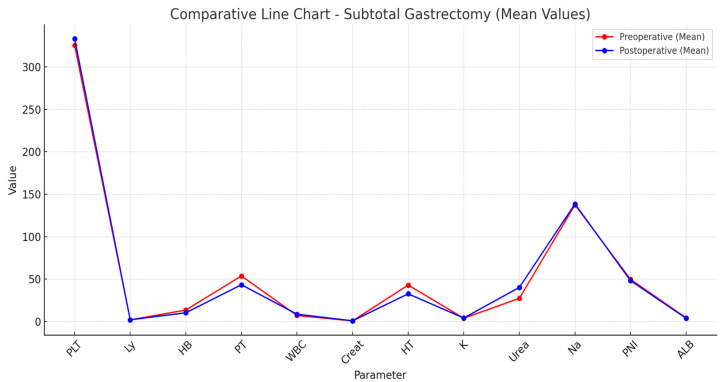
Comparative line chart. Subtotal Gastrectomy: ALB, serum albumin; Creat, creatinine; HB, hemoglobin; HT, hematocrit; K, potassium; Ly, lymphocytes; Na, sodium; PLT, platelets; PNI, prognostic nutritional index; PT, proteins; WBC, white blood count.

**Figure 3 nutrients-17-02146-f003:**
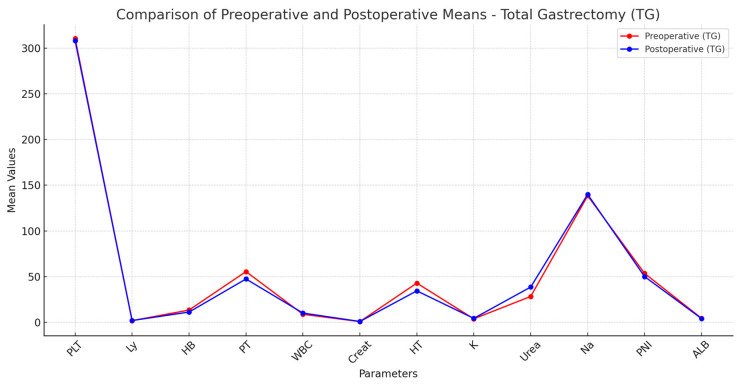
Comparative line chart. Total Gastrectomy: ALB, serum albumin; Creat, creatinine; HB, hemoglobin; HT, hematocrit; K, potassium; Ly, lymphocytes; Na, sodium; PLT, platelets; PNI, prognostic nutritional index; PT, proteins; TG, total gastrectomy; WBC, white blood count.

**Figure 4 nutrients-17-02146-f004:**
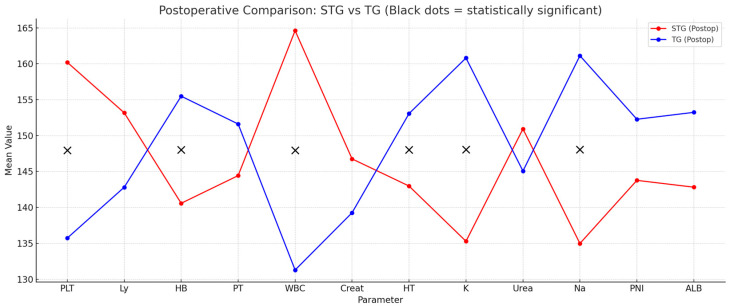
Postoperative comparison STG vs. TG: ALB, serum albumin; Creat, creatinine; HB, hemoglobin; HT, hematocrit; K, potassium; Ly, lymphocytes; Na, sodium; PLT, platelets; PNI, prognostic nutritional index; PT, proteins; STG, subtotal gastrectomy; TG, total gastrectomy; WBC, white blood count; X symbol indicates statistical significance.

**Table 1 nutrients-17-02146-t001:** Demographic and clinical characteristics of the study population (mean and percentage distribution). STG, subtotal gastrectomy; TG, total gastrectomy.

		Sex	Living Environment	Type of Resection	Number of Positive Lymph Nodes	Local Invasion	Age
	Description	295	295	295	295	295	295
Mean		0.59	0.56	0.5	8.3	0.5	64.9
Standard Deviation		0.5	0.4	0.5	11.6	0.4	10.4
Description	Feminine Rural areas STG-Distal location No invasion	123 (41.7%)	131 (44.4%)	145 (49.2%)		133 (45.1%)	
	Masculine Urban areas TG- Proximal location invasion	172 (58.3%)	164 (55.6%)	150 (50.8%)		162 (54.9%)	
Classification	Tubular Adenocarcinoma				123 (41.7%)		
	Papillary Adenocarcinoma				34 (11.5%)		
	Mucinous Adenocarcinoma				28 (9.5%)		
	Signet-Ring Cell Adenocarcinoma				23 (7.8%)		
	Mixed-Type Adenocarcinoma				87 (29.5%)		

**Table 2 nutrients-17-02146-t002:** Baseline normative data for biochemical and hematological markers. ALB, serum albumin; Creat, creatinine; HB, hemoglobin; HT, hematocrit; K, potassium; Ly, lymphocytes; Na, sodium; PLT, platelets; PT, proteins; WBC, white blood count.

Test Name	Reference Range
PLT	150,000–400,000 cellule/μL
Ly	1000–4800 cellule/μL
HB	F: 12–16/M: 13–17 g/dL
PT	6.4–8.3 g/dL
WBC	4000–10,000 cellule/μL
Creat	F: 0.6–1.1/M: 0.7–1.3 mg/dL
HT	F: 36–46/M: 40–50%
K	3.5–5.1 mmol/L
Urea	10–50 mg/dL
Na	135–145 mmol/L
ALB	3.5–5.0 g/dL

**Table 3 nutrients-17-02146-t003:** Descriptive statistics (mean) for biochemical parameters in patients undergoing STG preoperatively, *n* = 145, *p* > 0.05. ALB, serum albumin; Creat, creatinine; HB, hemoglobin; HT, hematocrit; K, potassium; Ly, lymphocytes; Na, sodium; PLT, platelets; PNI, prognostic nutritional index; PT, proteins; WBC, white blood count.

Subtotal Gastrectomy												
	**PLT**	**Ly**	**HB**	**PT**	**WBC**	**Creat**	**HT**	**K**	**Urea**	**Na**	**PNI**	**ALB**
Mean	325.4 ± 1.24	2.018 ± 0.98	13.6 ± 1.13	53.75 ± 0.89	7.2 ± 1.05	0.82 ± 1.31	43.02 ± 1.17	3.97 ± 0.94	27.38 ± 1.22	137.5 ± 1.08	49.75 ± 2.32	4.36 ± 1.15

**Table 4 nutrients-17-02146-t004:** Descriptive statistics (mean) for biochemical parameters in patients undergoing TG preoperative baseline values *n* = 150, *p* > 0.05. ALB, serum albumin; Creat, creatinine; HB, hemoglobin; HT, hematocrit; K, potassium; Ly, lymphocytes; Na, sodium; PLT, platelets; PNI, prognostic nutritional index; PT, proteins; WBC, white blood count.

Total Gastrectomy												
	**PLT**	**Ly**	**HB**	**PT**	**WBC**	**Creat**	**HT**	**K**	**Urea**	**Na**	**PNI**	**ALB**
Mean	310.4 ± 0.87	1.814 ± 1.02	13.4 ± 0.95	55.5 ± 1.1	8.8 ± 0.93	0.8 ± 1.09	42.9 ± 2.4	3.8 ± 1.3	28.3 ± 1.7	138.4 ± 1.12	53.5 ± 0.78	4.4 ± 1.12

**Table 5 nutrients-17-02146-t005:** Descriptive statistics for biochemical parameters in patients undergoing STG, post-operative baseline values *n* = 145. ALB, serum albumin; Creat, creatinine; HB, hemoglobin; HT, hematocrit; K, potassium; Ly, lymphocytes; Na, sodium; PLT, platelets; PNI, prognostic nutritional index; PT, proteins; WBC, white blood count.

Subtotal Gastrectomy												
	**PLT**	**Ly**	**HB**	**PT**	**WBC**	**Creat**	**HT**	**K**	**Urea**	**Na**	**PNI**	**ALB**
Mean	333.3 ± 9.4	2.112 ± 134.3	10.4 ± 0.2	43.4 ± 2.4	8.72 ± 0.7	0.89 ± 0.02	32.7 ± 0.6	4.2 ± 0.03	40.3 ± 1.19	138.6 ± 0.34	48.2 ± 0.99	3.9 ± 0.05
*p*	0.05	0.3	0.01	0.2	0.2	0.1	0.02	0.02	0.3	0.005	0.1	0.1

**Table 6 nutrients-17-02146-t006:** Descriptive statistics for biochemical parameters in patients undergoing TG, postoperative baseline values *n* = 150. ALB, serum albumin; Creat, creatinine; HB, hemoglobin; HT, hematocrit; K, potassium; Ly, lymphocytes; Na, sodium; PLT, platelets; PNI, prognostic nutritional index; PT, proteins; WBC, white blood count.

Total Gastrectomy												
	**PLT**	**Ly**	**HB**	**PT**	**WBC**	**Creat**	**HT**	**K**	**Urea**	**Na**	**PNI**	**ALB**
Mean	308.01 ± 9.1	1.962 ± 86.4	11.2 ± 0.19	47.4 ± 2.4	10.1 ± 1.12	0.9 ± 0.05	34.4 ± 0.49	4.3 ± 0.03	38.6 ± 1.17	139.9 ± 0.28	50.01 ± 0.85	4.07 ± 0.049
*p*	0.05	0.3	0.01	0.2	0.2	0.1	0.02	0.02	0.3	0.005	0.1	0.1

**Table 7 nutrients-17-02146-t007:** Comparative Descriptive Statistics of Biochemical Parameters in Patients Undergoing TG versus STG (*n* = 295). ALB, serum albumin; Creat, creatinine; HB, hemoglobin; HT, hematocrit; K, potassium; Ly, lymphocytes; Na, sodium; PLT, platelets; PNI, prognostic nutritional index; PT, proteins; STG, subtotal gastrectomy; TG, total gastrectomy; WBC, white blood count.

Subtotal Gastrectomy vs. Total Gastrectomy												
	PLT	Ly	HB	PT	WBC	Creat	HT	K	Urea	Na	PNI	ALB
Mean STG	160.2	153.17	140.57	144.42	164.61	146.75	142.97	135.28	150.92	134.97	143.76	142.81
Mean TG	135.72	142.8	155.48	151.61	131.28	139.25	153.06	160.8	145.06	161.12	152.27	153.23
Mann Whitney U	9.073	10.113	9.779	10.348	8.419	9.978	10.134	8.996	10.445	8.948	10.250	10.109
*p*	0.01	0.2	0.01	0.4	0.001	0.1	0.01	0.01	0.3	0.008	0.3	0.2

## Data Availability

The original contributions presented in this study are included in the article. Further inquiries can be directed to the corresponding author.

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
