# Peer review of "Nutritional and Biochemical Outcomes After Total Versus Subtotal Gastrectomy: Insights into Early Postoperative Prognosis"

_nutrients, 2025, doi:10.3390/nu17132146_

Round 1

Reviewer 1 Report (Previous Reviewer 1)

Comments and Suggestions for Authors

Our colleagues propose us again the paper on gastric cancer reworked, the abstract is a summary in which the purpose of the study does not appear, which should be well explained. The introduction should better frame the disease they want to write about and only in the final lines does the aim of their work appear. In the materials and methods the cornerstones of the diagnostic/therapeutic path should be better explained. Was a diagnosis made on the biopsy sample regarding microbiology? Satellites and HER 2? Furthermore, no mention is made of the discussion of cases in a multidisciplinary commission and of the patient's taking charge by a nutritionist (doi.org/10.3390/nu17010188 to be read and reported in the bibliography). Also important are the criteria for establishing whether the patient should be started on neoadjuvant therapy or up-front surgery. In the postoperative period, what rules did the follow-up follow for adjuvant therapy? In our opinion, colleagues should revise the paper following the advice given.

Author Response

colleagues propose us again the paper on gastric cancer reworked,

  1. The abstract is a summary in which the purpose of the study does not appear, which should be well explained.
  • The abstract has been revised to include a clear and early statement of the study objective, highlighting the purpose of evaluating early postoperative biochemical and nutritional differences between total and subtotal gastrectomy. The revised manuscript reflects these clarifications (lines 23–42). See the modification here nutrients%20VM%2024Mai%20modificat.docx#Abstract

  1. The introduction should better frame the disease they want to write about and only in the final lines does the aim of their work appear.
  • We have revised the introduction to provide a more comprehensive background on gastric cancer, including its global epidemiology, prognostic implications, and nutritional challenges. However, we would like to clarify that the primary focus of this study is not on the histological subtypes of gastric cancer, but rather on the type of surgical intervention (total vs. subtotal gastrectomy) and its early postoperative impact on biochemical and nutritional parameters. Therefore, the introduction was strengthened to reflect the clinical relevance of these surgical approaches while maintaining the study’s primary focus. The revised manuscript reflects these clarifications (lines 50–59 and 79-107). See the modification here nutrients%20VM%2024Mai%20modificat.docx#GC

  • The objective of the study has been moved earlier in the introduction to improve clarity and guide the reader more effectively. The revised paragraph now explicitly states the aim of the study within the first section of the introduction, immediately following the contextual background on gastric cancer and its nutritional implications. The revised manuscript reflects these clarifications (lines 60-65). See the modification herenutrients%20VM%2024Mai%20modificat.docx#Inthisstudy

  1. In the materials and methods the cornerstones of the diagnostic/therapeutic path should be better explained.
  • We have revised the Materials and Methods section to provide a more comprehensive description of the diagnostic and therapeutic pathway. Specifically, we clarified that all patients were diagnosed via endoscopic biopsy, underwent preoperative staging with contrast-enhanced thoraco-abdominal CT, and were evaluated by a multidisciplinary tumor board to determine the appropriate treatment strategy. Patients who had received neoadjuvant therapy were excluded from the analysis in order to isolate the effect of surgical technique on early postoperative biochemical and nutritional outcomes. We also specified the criteria used to select patients for upfront surgery (clinical stage T1–T3N0–N1, ECOG 0–1, no distant metastasis), and detailed the rationale for choosing total versus subtotal gastrectomy based on tumor location and extent. These additions are now included in the revised manuscript (lines 112–130). See the modification here nutrients%20VM%2024Mai%20modificat.docx#Thepatients

  1. Was a diagnosis made on the biopsy sample regarding microbiology? Satellites and HER 2?
  • Microbiological or molecular characterization (including Helicobacter pylori detection, HER2 status, or microsatellite instability) was not included in the diagnostic protocol analyzed in this study, as the primary objective was to evaluate the early postoperative biochemical and nutritional changes associated with the type of gastrectomy

  1. Furthermore, no mention is made of the discussion of cases in a multidisciplinary commission and of the patient's taking charge by a nutritionist (doi.org/10.3390/nu17010188 to be read and reported in the bibliography).
  • We confirm that all patients received nutritional assessment and monitoring during hospitalization under the supervision of the surgical team, and, in cases of significant nutritional deficits, it was noted in the revised manuscript lines 113-121 and 434-436. See the modification here nutrients%20VM%2024Mai%20modificat.docx#Allcases and here nutrients%20VM%2024Mai%20modificat.docx#Nutritionalsupport

  1. Also important are the criteria for establishing whether the patient should be started on neoadjuvant therapy or up-front surgery. In the postoperative period, what rules did the follow-up follow for adjuvant therapy?
  • As clarified in the revised Materials and Methods section, only patients who were treated with upfront surgery were included in the present study. The selection was based on clinical staging (T1–T3, N0–N1, M0), performance status (ECOG 0–1), and multidisciplinary tumor board recommendations, in line with current ESMO and NCCN guidelines. Regarding the postoperative period, we have added a paragraph to describe the follow-up protocol. All patients underwent complete histopathological staging after surgery. Those with stage II or III disease, lymph node involvement, or other high-risk features were recommended for adjuvant chemotherapy based on tumor board decisions. These patients were referred to oncology and followed through structured protocols including periodic imaging, tumor marker monitoring, and clinical evaluations. The revised manuscript reflects these clarifications (lines XX–XX). These additions are now included in the revised manuscript (lines 112–130). See the modification here nutrients%20VM%2024Mai%20modificat.docx#Thepatients

Reviewer 2 Report (New Reviewer)

Comments and Suggestions for Authors

nutrients-3691845

Type of manuscript: Article
Title: Nutritional and biochemical outcomes after total versus subtotal gastrectomy: Early postoperative implications for prognostic assessment
Authors: Fawzy Akad, Cristinel Ionel Stan *, Florin Zugun-Eloae, Sorin Nicolae Peiu, Nada Akad, Dragos-Valentin Crauciuc, Marius Constantin Moraru, Cosmin Gabriel Popa, Liviu-Ciprian Gavril, Roxana-Florentina Sufaru, Cristina Preda, Veronica Mocanu *

This manuscript addresses an important and clinically relevant topic: the early postoperative nutritional and biochemical consequences of total versus subtotal gastrectomy in patients with gastric cancer. The study's prospective design, reasonably large cohort size, and comprehensive laboratory assessments add value to its findings. The comparison between the two surgical approaches in the early postoperative period is particularly timely and has direct implications for improving patient outcomes through tailored postoperative care.

However, I think further improvements are needed in the following areas:

[Major concerns]

  1. Lack of long-term follow-up: While the focus is on early postoperative outcomes (day 14), additional insights from 1-month or 3-month follow-up would help evaluate whether these disturbances persist, normalize, or worsen over time. Therefore, it would be helpful for the authors to clarify their position on studies involving long-term follow-up.
  2. Absence of functional or clinical outcome measures: The study could be strengthened by correlating laboratory abnormalities with clinical endpoints, such as length of hospital stay, postoperative complications, or patient-reported outcomes. Therefore, consider integrating clinical outcome data to better connect laboratory findings with patient prognosis.
  3. Discussion depth: While the findings are clearly presented, the discussion lacks an in-depth exploration of possible pathophysiological mechanisms behind the observed differences and how these align or differ from existing literature. Enhance the discussion section by referencing current guidelines and explaining the observed gender-related differences.
  4. Nutritional management details: The manuscript recommends personalized nutritional strategies but does not describe any specific interventions used in the cohort or propose structured management protocols.
  5. Add a supplementary table or figure to visually compare pre- and postoperative parameter changes across subgroups.
  6. Types of gastric adenocarcinomas: In the Materials and Methods section, five types of gastric cancer are listed in the following order: tubular adenocarcinoma (AC), papillary AC, mucinous AC, signet ring cell AC, and mixed-type AC. However, only the types are listed. There may be statistical data on their worldwide incidence, so please add this part. In addition, when listing two or more of these five types of gastric cancer in the results of this study, if possible, always list them in a certain order so that readers can easily understand them.
  7. Abbreviations: The use of abbreviations when writing a paper has many advantages besides simplicity of expression. To use an abbreviation, first write the abbreviation in parentheses after the full name, and then use the abbreviation from Introduction to the final Conclusion. Abbreviations should only be used if they are repeatedly used and if they are not used again, only the full name should be used. Abbreviations should be written separately in the abstract and the main text. Since there are cases where the abstract is introduced alone, abbreviations used in the abstract should be used only when they are repeated in the abstract. Whether it's an abstract or a text, systematically proofread each abbreviation to make sure there are no abbreviations that have been redefined repeatedly.
  8. In cases where abbreviations are used within figures or tables, please list these abbreviations along with their corresponding full names in the figure legends or at the bottom of corresponding tables. If there are two or more abbreviations, arrange them in alphabetical order. In this case, non-proper nouns should not have their first letters capitalized. And when listing abbreviations and full names, rewrite them according to the following examples. Examples: ALB, serum albumin; Creat, creatinine; HB, hemoglobin; HT, hematocrit; K, potassium; Ly, lymphocytes; Na, sodium; PLT, platelets; PT, proteins; WBC, white blood count.
  9. ‘CA19-9’ at Line 56 vs.’ CA 19-9’ at Line 210; etc.: The notation of major keywords should always be consistent. Please check the entire paper again to see if there are any similar cases to this one.
  10. In this paper, the notation of numerical values with decimal points is inconsistent, with both the international standard and the method commonly used in Europe being used. It would be preferable to unify the notation according to the international standard, in line with the conventions of international academic journals. For example: ‘53.3’ vs. ‘53,3’.
  11. Tables 3–7: A wide range of biochemical parameters are listed in the tables, but their units are not indicated at all. Please provide appropriate units for each parameter."

[Minor concerns]

  1. Line 87: Define WHO.
  2. Line 119 ‘(a jeun)’: It's an English paper, so is there really a need to put French in parentheses?
  3. Line 165: Arabic numerals should not appear as the first word in an English sentence. Please make the necessary corrections accordingly.

Overall, the manuscript can be considered to publication after major revision as indicated above.

Author Response

[Major concerns]

  1. Lack of long-term follow-up: While the focus is on early postoperative outcomes (day 14), additional insights from 1-month or 3-month follow-up would help evaluate whether these disturbances persist, normalize, or worsen over time. Therefore, it would be helpful for the authors to clarify their position on studies involving long-term follow-up.
  • The study was intentionally designed to focus on the acute postoperative phase, with a 14-day follow-up period, in order to capture early biochemical and hematological dynamics that are most relevant for immediate clinical care and complication monitoring. While we acknowledge the importance of long-term data, such as 1- or 3-month follow-up, this was beyond the scope of the present study. We have clarified this design choice in the Discussion section (lines 511–536), and we recognize it as a limitation that opens the way for future longitudinal research. See the modification here nutrients%20VM%2024Mai%20modificat.docx#Thisstudywasdesigned

  1. Absence of functional or clinical outcome measures: The study could be strengthened by correlating laboratory abnormalities with clinical endpoints, such as length of hospital stay, postoperative complications, or patient-reported outcomes. Therefore, consider integrating clinical outcome data to better connect laboratory findings with patient prognosis.
  • In our institutional setting, patients remained hospitalized for a median duration of 14 days and were discharged only upon achieving clinical stabilization and normalization of key biochemical parameters. Although formal clinical outcome measures, such as length of stay, postoperative complications, or patient-reported outcomes were not independently analyzed, the discharge criteria themselves provide an implicit clinical endpoint grounded in biochemical recovery. This approach offered a uniform benchmark for evaluating early postoperative status. We have added a clarifying statement in the Discussion section (lines 511–536), and we recognize it as a limitation that opens the way for future longitudinal research. See the modification here nutrients%20VM%2024Mai%20modificat.docx#Thisstudywasdesigned

  1. Discussion depth: While the findings are clearly presented, the discussion lacks an in-depth exploration of possible pathophysiological mechanisms behind the observed differences and how these align or differ from existing literature. Enhance the discussion section by referencing current guidelines and explaining the observed gender-related differences.
  • Although our analysis did not reveal statistically significant differences between male and female patients, and the study was not specifically designed to explore gender as a variable, we acknowledge the biological plausibility and clinical relevance of such differences, as supported by existing literature. Additionally, the distribution between male and female patients was not balanced across the two surgical subtypes (TG vs. SG), which further limited the interpretability of any gender-based comparison. Therefore, we decided not to focus the analysis on sex-related outcomes. Nevertheless, to address this important point, we have included a supplementary discussion (lines 459–486) summarizing current evidence on gender-related immune modulation in the postoperative context and proposing this topic as a valuable direction for future studies. See the modification here nutrients%20VM%2024Mai%20modificat.docx#Surgicaltrauma

  1. Nutritional management details: The manuscript recommends personalized nutritional strategies but does not describe any specific interventions used in the cohort or propose structured management protocols.
  • While no formal or standardized nutritional protocol was implemented across the cohort, supportive nutritional care was provided throughout the 14-day postoperative period in line with clinical needs. Electrolyte imbalances, such as hypokalemia, were corrected as necessary, and for patients presenting with significant hypoalbuminemia or protein deficits, intravenous protein or albumin supplementation (e.g., human serum albumin solutions) was administered based on clinical indications. These interventions were guided by individual laboratory profiles and physician judgment, rather than a structured nutritional algorithm. We have clarified this approach in the revised manuscript (lines 434–443), we aim to emphasize the need for a well-defined and structured nutritional management protocol, as our findings highlight the metabolic vulnerabilities that arise in the early postoperative period and the potential benefit of standardized interventions. See the modification here nutrients%20VM%2024Mai%20modificat.docx#Nutritionalsupport

  1. Add a supplementary table or figure to visually compare pre- and postoperative parameter changes across subgroups.
  • As requested, three supplementary clustered column charts were added to visually compare pre- and postoperative changes in key parameters across subgroups. These visualizations enhance the clarity of data interpretation and highlight significant trends in a reader-friendly format. The corresponding modifications can be found in the manuscript at lines 323–325, 342–345, and 358–362. See the modification here nutrients%20VM%2024Mai%20modificat.docx#Figure2, nutrients%20VM%2024Mai%20modificat.docx#Figure3, nutrients%20VM%2024Mai%20modificat.docx#Figure4

  1. Types of gastric adenocarcinomas: In the Materials and Methods section, five types of gastric cancer are listed in the following order: tubular adenocarcinoma (AC), papillary AC, mucinous AC, signet ring cell AC, and mixed-type AC. However, only the types are listed. There may be statistical data on their worldwide incidence, so please add this part. In addition, when listing two or more of these five types of gastric cancer in the results of this study, if possible, always list them in a certain order so that readers can easily understand them.
  • As requested, a new paragraph was added (lines 163–166) including global incidence data for the five histological subtypes of gastric adenocarcinoma, based on WHO and large-scale pathology studies. To ensure consistency and improve clarity for the reader, the predefined order of subtypes: tubular, papillary, mucinous, signet-ring cell, and mixed-type adenocarcinoma has been applied uniformly throughout the manuscript. Accordingly, the paragraph between lines 260–269 was revised to reflect this standardized order. See the modification here nutrients%20VM%2024Mai%20modificat.docx#Globally and nutrients%20VM%2024Mai%20modificat.docx#Regarding

  1. Abbreviations: The use of abbreviations when writing a paper has many advantages besides simplicity of expression. To use an abbreviation, first write the abbreviation in parentheses after the full name, and then use the abbreviation from Introduction to the final Conclusion. Abbreviations should only be used if they are repeatedly used and if they are not used again, only the full name should be used. Abbreviations should be written separately in the abstract and the main text. Since there are cases where the abstract is introduced alone, abbreviations used in the abstract should be used only when they are repeated in the abstract. Whether it's an abstract or a text, systematically proofread each abbreviation to make sure there are no abbreviations that have been redefined repeatedly.
  • All abbreviations throughout the manuscript have been reviewed and updated according to the required guidelines. Repeated definitions and inconsistencies have been corrected, both in the abstract and main text. Abbreviations are now introduced only once (at first mention) and used consistently thereafter. In tables and figures, all abbreviations are listed in alphabetical order with their corresponding full terms provided in the legends or footnotes, following the recommended formatting.

  1. In cases where abbreviations are used within figures or tables, please list these abbreviations along with their corresponding full names in the figure legends or at the bottom of corresponding tables. If there are two or more abbreviations, arrange them in alphabetical order. In this case, non-proper nouns should not have their first letters capitalized. And when listing abbreviations and full names, rewrite them according to the following examples. Examples: ALB, serum albumin; Creat, creatinine; HB, hemoglobin; HT, hematocrit; K, potassium; Ly, lymphocytes; Na, sodium; PLT, platelets; PT, proteins; WBC, white blood count.
  • The requested changes regarding the formatting and listing of abbreviations have been implemented in all tables. Each abbreviation is now followed by its corresponding full term, arranged in alphabetical order in the table footnotes. The rest of the manuscript was carefully reviewed, and no additional abbreviation-related issues were identified outside the tables.

  1. ‘CA19-9’ at Line 56 vs.’ CA 19-9’ at Line 210; etc.: The notation of major keywords should always be consistent. Please check the entire paper again to see if there are any similar cases to this one.
  • The spacing inconsistency in “CA19-9” vs. “CA 19-9” has been corrected. The entire manuscript was carefully reviewed for similar formatting issues involving abbreviations and key terms, and no other inconsistencies were found.
  1. In this paper, the notation of numerical values with decimal points is inconsistent, with both the international standard and the method commonly used in Europe being used. It would be preferable to unify the notation according to the international standard, in line with the conventions of international academic journals. For example: ‘53.3’ vs. ‘53,3’.
  • All numerical values have been reviewed and standardized according to international formatting conventions, using a decimal point (e.g., 53.3 instead of 53,3). The entire manuscript was checked and revised.
  1. Tables 3–7: A wide range of biochemical parameters are listed in the tables, but their units are not indicated at all. Please provide appropriate units for each parameter."
  • A reference table containing the baseline normative ranges for biochemical and hematological markers, along with their respective measurement units, has been added to the Evaluation of biochemical profiles section (Materials and Methods, lines 197–199). This addition ensures clarity and facilitates interpretation by providing standardized reference values for all reported parameters.See the modification here nutrients%20VM%2024Mai%20modificat.docx#Table2

[Minor concerns]

  1. Line 87: Define WHO.
  • We have now defined the abbreviation “WHO” as “World Health Organization” at its first mention in the manuscript (line 149 after the modifications were made), in accordance with standard scientific writing conventions. See the modification here nutrients%20VM%2024Mai%20modificat.docx#Histopathologicalanalysis
  1. Line 119 ‘(a jeun)’: It's an English paper, so is there really a need to put French in parentheses?
  • The French word has been removed to maintain linguistic consistency throughout the paper.
  1. Line 165: Arabic numerals should not appear as the first word in an English sentence. Please make the necessary corrections accordingly.
  • All necessary corrections have been made to ensure that no sentence begins with Arabic numerals.

Round 2

Reviewer 1 Report (Previous Reviewer 1)

Comments and Suggestions for Authors

We have carefully read the changes made to the paper by the authors on the article they produced. The initial part has been substantially changed and some sentences have also been changed in the rest of the article that have not changed it in substance but have made it concrete in the key concepts. Endorsement publication

This manuscript is a resubmission of an earlier submission. The following is a list of the peer review reports and author responses from that submission.

Round 1

Reviewer 1 Report

Comments and Suggestions for Authors

Interesting study on gastric cancer. The study is retrospective and involved a fairly significant number, 295, of patients, in a short period of time. In our opinion, it is appropriate to review the paper from the introduction, clarifying better what is the aim of the same. Furthermore, from this, it must be clear that today the surgical act is no longer the cornerstone of the treatment of this disease, which still represents one of the main causes of death by neoplasia in the world. At the moment the treatment is based on four cardinal points that allow us to frame the pathology and personalize diagnosis and treatment. The first phase is the diagnosis and we have the endoscopy available that will perform biopsies on which the pathologists will be able to provide us with the grade (G1, G2, G3) but also the HER2, which can be positive or negative; but it will also be possible to know if the microsatellites will be wild or not. This will already allow us to frame the pathology that we will find ourselves facing and to understand which chemotherapy treatment will be most useful. Imaging will therefore be essential for the staging of any lymph node stations involved and metastases due to contiguity and/or distance. The doubt that the neoplasia is surfacing or there will be the presence of peritoneal fluid will be necessary to schedule an exploratory laparoscopy to evaluate the possible PCI. Neoplastic markers are routinely done on everyone, but we know that in the stomach these values ​​are not very significant, as the authors themselves say in the text they wrote, so I would dedicate less space to them. At this point we have the second phase because we have enough elements to discuss the case in a multidisciplinary commission to undertake or not a neoadjuvant treatment based on, as per the guidelines on FLOT (DOI: 10.1016/S0140-6736(18)32557-19 to be read and cited in the bibliography) we remember that the patient must be followed by a nutritionist from the early stages of treatment to avoid that lowering of metabolic parameters mentioned in the text. After 4/6 cycles of FLOT and the control CT, we can also plan a treatment with PIPAC for any peritoneal lesions. The case must then be submitted to the multidisciplinary commission to decide whether it is time for surgery. The third phase is precisely the surgical one and colleagues write to us that their patients have undergone gastric resection and recommend lymphadenectomy, but they must specify that this can be D2, D2 plus (doi.org/10.3390/cancers16071376 to be read and cited in the bibliography) or D3. The definitive histology must therefore present a number of lymph nodes greater than 16 according to international guidelines, but the number is still considered limited. After about a month of rehabilitation treatments, in which the main one is nutritional, the patient must be started on the third phase, that is, adjuvant therapy and in case of advanced disease the therapy must initially be advanced chemotherapy (DOI: 10.1097/CAD.000000000000000877 to be read and cited in the bibliography). Colleagues write a discussion in light of the results produced by their database and what they have correctly elaborated with the consulted bibliography and with their arguments, but I would revise the article giving less importance to neoplastic markers and basing their database on the reflections arising from this review. In reality we can already say that we are on the road to personalizing this disease. Good English, good bibliography.

Author Response

Thank you very much for your time and thoughtful comments. Your observations have been extremely helpful in improving the focus and clarity of our manuscript.

We would like to take this opportunity to clarify the primary objective and scope of our study. Although the context is that of patients undergoing surgery for gastric cancer, the core aim of the manuscript is not to assess oncological management pathways, but rather to evaluate the early postoperative metabolic and nutritional impact of different types of gastrectomy. Our interest lies in determining whether subtotal versus total gastrectomy leads to more significant deficits in hematological and biochemical markers and, implicitly, in nutritional status.

The inclusion of oncologic data, such as tumor location, histopathological staging, lymph node involvement, and even tumor markers (CEA and CA 19-9), was not intended as a focal point of analysis. These variables were presented to offer clinical context, to document how surgical decisions were made, and to stratify the patient cohort accordingly. As you rightly pointed out, these aspects reflect a modern and multidisciplinary approach to gastric cancer, but in the present study they are used primarily to define the background and do not constitute endpoints.

Following your suggestions, we revised the manuscript in several key areas:

  • We clarified that tumor markers were excluded from detailed analysis due to lack of consistent follow-up data across the cohort. However, a brief note remains in the discussion, indicating a downward trend postoperatively in most cases, which may suggest a future role for these markers in monitoring recurrence. The full data have been removed to maintain scientific rigor.
  • We significantly expanded the explanation of the timing and relevance of postoperative data collection, which was uniformly conducted at two weeks post-surgery. This specific interval was chosen because it provided the most complete laboratory dataset and corresponded with a phase in which metabolic and nutritional deterioration begins to manifest clearly. By observing these early dynamics, we aimed to identify which surgical group (subtotal or total gastrectomy) presents the greatest clinical risk and may therefore benefit from earlier and more intensive nutritional interventions.
  • Although we do not include a placebo group or adjuvant chemotherapy comparison, the goal was to extract meaningful intra-group differences, in order to support a clinical model of anticipatory care. This approach may eventually contribute to adjusting perioperative nutritional strategies based on surgical type, with the aim of preventing protein-calorie malnutrition and optimizing recovery.

We fully agree with your observation that current gastric cancer management relies on a complex, multidisciplinary framework including neoadjuvant chemotherapy (FLOT protocol), molecular profiling (HER2, MSI), and nutritional monitoring from early stages. These components are indeed essential in clinical practice and were referenced in our revised introduction for context, but our study remains centered on nutritional outcomes, not therapeutic algorithms.

To better reflect this focus, we have reformulated the title and revised the introduction and discussion sections accordingly. We hope these changes now better align the manuscript with its intended scientific contribution and address your insightful observations.

Reviewer 2 Report

Comments and Suggestions for Authors

please see the attached document.

Author Response

would like to sincerely thank you for your valuable feedback regarding our manuscript. We have carefully addressed each of your comments and revised the text accordingly. Below is a summary of the changes made in response to your specific points:

  1. Tumor location and surgical decision-making

The paragraph addressing this aspect was  now updated and expanded it to provide greater clarity and detail. The revised version clarifies that the type of gastrectomy was selected in accordance with established surgical guidelines, based on tumor location, local invasion, and oncologic criteria. Tumor localization was determined preoperatively using upper gastrointestinal endoscopy and contrast-enhanced CT. These changes can be found in Lines 71–80 of the revised manuscript and are now supported by references [19] and [20].

  1. Use of postoperative pathological data

We have provided a more detailed explanation regarding the role of both preoperative and postoperative data.  Preoperative data were indeed collected for all patients, but in the initial version of the study, these values were not included, as they fell within normal reference ranges and did not show statistically significant variation. For clarity and to maintain focus on the relevant clinical changes, these parameters were initially omitted. However, in the revised version of the actual manuscript, we have included the preoperative data to provide a more complete overview and ensure transparency of the methodology.

Instead, we concentrated on postoperative pathological parameters, specifically histopathological features, lymph node involvement, and local invasion status, which were analyzed retrospectively. These three variables were not used to guide the initial surgical approach, but rather served a dual purpose: (1) to support and validate the appropriateness of the selected surgical intervention in each case, and (2) to allow for better stratification and classification of the patient cohort based on the severity and extent of disease.

This analytical focus enabled a more coherent presentation of the surgical outcomes, eliminating redundant or non-contributory information that did not add value to the clinical interpretation. In addition, we clarified the timing of postoperative data collection, which was standardized to approximately two weeks after surgery, as this interval provided the most consistent biochemical and hematological profiles across patients.

These updates and clarifications can now be found in Lines 117–124 and 217–281 of the revised manuscript.

  1. Role of tumor markers (CEA and CA 19-9)

Following your suggestion, we have significantly revised this section. After further evaluation, we decided to remove the detailed data regarding tumor markers CEA and CA 19-9, which were initially included. Although these markers were measured both preoperatively and postoperatively, we were unable to ensure long-term follow-up data for the entire patient cohort. Consequently, to avoid presenting incomplete or potentially misleading results, we chose to exclude the quantitative analysis of these markers from the revised version of the study.

Instead, we now include a concise statement noting that a postoperative decrease in CEA and CA 19-9 levels was observed in most patients, suggesting their potential relevance in tracking tumor dynamics and response to surgery. However, due to the lack of longitudinal consistency, this observation remains exploratory. These changes and clarifications are now reflected in Lines 324–336 of the revised study.

  1. Nutritional and metabolic evaluation without follow-up

We explained that the study’s aim was to compare early postoperative metabolic and nutritional status between subtotal and total gastrectomy, not to analyze individual pre- and postoperative progression. We clarified that the analysis incorporates both preoperative and postoperative (day 14) data, in order to capture the early dynamics of biochemical, hematological, and nutritional changes following surgery. The aim was to observe whether these early modifications could be correlated with alterations in nutritional status, and to determine if there is a statistically significant difference between the subtotal and total gastrectomy groups. This comparison helps identify which group is more vulnerable in the immediate postoperative period and may therefore benefit from early nutritional intervention and closer monitoring, to prevent malnutrition that could potentially interfere with the administration of future therapies. These clarifications have been incorporated into the discussion section, Lines 367–399, of the revised manuscript.

  1. As part of the manuscript revision process, we have decided to reformulate the original title. The initial version did not fully reflect the core message and objectives of the study. Following the structural and content-related updates, the title was revised to more accurately highlight both the scope of the research and the main findings obtained. The new title provides a clearer representation of the study’s focus on the postoperative metabolic and nutritional dynamics in relation to the type of gastrectomy performed.

Please let us know if any further clarification is needed. We appreciate your thoughtful review and the opportunity to improve our manuscript.
